# Enrichment Is Simple, That’s the Problem: Using Outcome-Based Husbandry to Shift from Enrichment to Experience

**DOI:** 10.3390/ani12101293

**Published:** 2022-05-18

**Authors:** Greg A. Vicino, Jessica J. Sheftel, Louisa M. Radosevich

**Affiliations:** San Diego Zoo Wildlife Alliance, San Diego, CA 92101, USA; jsheftel@sdzwa.org (J.J.S.); lradosevich@sdzwa.org (L.M.R.)

**Keywords:** zoo, behavior, welfare, experience, enrichment, husbandry, leopard

## Abstract

**Simple Summary:**

As animal care practitioners continue to advance husbandry practices, traditional methodologies are continually being reevaluated with a critical eye on efficacy. When evaluating practices designed to maximize wildlife care and welfare, environmental enrichment remains one of the most well-documented and deployed strategies in the managed care of wildlife. Enrichment does, however, have limitations and is most often considered a supplemental component of animal care. It is the supplemental nature of traditional enrichment that lends itself to being overly dependent on inputs from caretakers and lacks relevance to the natural history of the species. By utilizing a tool to highlight relevant outcomes when designing husbandry programs, it is our position that animals in managed care can have a more complete experience that is relative to their adaptations. The provisioning of resources, facilitation of self-maintenance, and care programs that require animal-driven choices may be able to dispel the notion that enrichment is required to augment typical animal care.

**Abstract:**

Over the decades, the use of environmental enrichment has evolved from a necessary treatment to a “best practice” in virtually all wildlife care settings. The breadth of this evolution has widened to include more complex inputs, comprehensive evaluation of efficacy, and countless commercially available products designed to provide for a myriad of species-typical needs. Environmental enrichment, however, remains almost inexorably based on the provision of inputs (objects, manipulanda, or other sensory stimuli) intended to enhance an environment or prolong a specific behavior. Considerable effort has been put into developing enrichment strategies based on behavioral outcomes to shift the paradigm from the traditional input-heavy process. We believe that this trajectory can be enhanced through Outcome-Based Husbandry using an ethologically based workflow tool with a universal application (regardless of species) that flushes out inputs based on desired outcomes, which can then be incorporated into daily care or layered to create sensory cue-based multi-day events. Furthermore, we believe that this strategy can drive practitioners from the confines of traditional enrichment and the object-based approach into a dynamic and holistic husbandry program that synthesizes complex experiences into regular animal care, rather than supplementing husbandry with input-based enrichment. Focusing on an animal’s complete experience and outcomes that promote competence building and the highest level of agency allows the animals, not care staff, to make meaningful decisions that impact their present and future selves.

## 1. Introduction

Referring to the area where an animal resides as a habitat instead of a cage, enclosure, or exhibit, regardless of actual physical structure, can influence basic perceptions of animal living standards [1]. Words are designed to communicate meaning but can drift and evolve based on how they are employed, no longer representative of the original intention. It is especially true when a word (or term) defines specific practices in a field as specialized and nuanced as the care of wildlife in a zoo, aquarium, sanctuary, or pre-release facility. The word “enrichment” is one of those terms, and even while enrichment programs have tried to evolve from a focus on items to a behavior-based approach to husbandry [2], the term enrichment is still defined by most caretakers as any object inserted into a habitat that activates the animal(s) in some way. Enrichment has lost its meaning, and relying on enrichment as the “thing” that makes the animals’ lives better hinders other opportunities to enhance more holistic husbandry efforts. The history and breadth of environmental enrichment in the literature and practice is extensive [3,4], interspersed with some notable paradigm-shifting breakthroughs, such as interactive devices, biologically and ethologically relevant programs, and the formal S.P.I.D.E.R model (Setting goals, Planning, Implementation, Documentation, Evaluation, Re-Adjustment) to measure efficacy [5,6,7,8] There are also some truly consequential theories to help foster the concept of an animal’s agency and the role that challenge plays in fostering a well-rounded mental state for managed animals [9,10,11]. These examples are by no means exhaustive, but they do serve as the representative body that anchors most modern environmental enrichment programs and practices. They will also serve as the jumping-off point to examine the true nature of how animals in managed care experience the world. This paper seeks to build on the body of literature by outlining the process of developing an Outcome-Based Husbandry (OBH) program that shifts the focus of enrichment from an input-based program to one that enhances wildlife ability to use their skills and relevant behaviors to engage with their environment. 

## 2. Enrichment ‘Can’ Provide Experiences

A somewhat consistent theme of most modern enrichment programs remains the supplemental nature of the enrichment itself, and the input-driven approach to application. Theoretically, this may be due to the very compliance requirements in place to ensure enrichment is provided to wildlife in human care. For example, in the U.S., the Animal and Plant Health Inspection Service (APHIS) under the United States Department of Agriculture (USDA) enforces the Animal Welfare Act [12] by providing specific Animal Welfare Regulations outlined in what is colloquially known as the “blue book” [13]. The regulations include a very specific section outlining that enrichment is to be provided for all non-human primates, stating, “The physical environment in the primary enclosures must be enriched by providing means of expressing noninjurious species-typical activities” (USDA, Subpart D, 3.81 (b), 2019). Subpart D, Section 3.81 (b) of the regulations also provides examples of appropriate enrichment: “Examples of environmental enrichments include providing perches, swings, mirrors, and other increased cage complexities; providing objects to manipulate; varied food items; using foraging or task-oriented feeding methods; and providing interaction with the caregiver or other familiar and knowledgeable person consistent with personnel safety precautions.” The focus is solely directed at inputs and objects, with no clear guidance on the appropriate outcome, barring the stipulation that any enrichment must “provide means of expressing non-injurious species-typical activities.” Such definitions are common in compliance documents related to enrichment, but they exacerbate the acceptance of input-driven enrichment models as the norm. This example also highlights the clear distinction between enrichment and husbandry, further isolating enrichment as a supplement to normal wildlife care. 

Another critical piece missing from traditional enrichment programs is the adaptive value of the behavior expressed by the animal following the provision of supplemental enrichment. The term “species-specific behavior” is the foundational goal of enrichment; however, as eloquently detailed by Mellen and MacPhee [7], the term may not be so straightforward. Species-specific behaviors typically evoke images of a particular species’ behaviors as they would be seen in the wild or natural environment. Behaviors like infanticide, predation, or fatal wounding due to competition would likely not be appropriate in managed care [14]. Using this logic, one could argue that it is the frequency and diversity of behavior that is appropriate for any given species, within the given context that it finds itself [15]. Furthermore, most enrichment programs follow a typical pattern of recording the impact of the enrichment (in terms of behavioral output), with varying degrees of accuracy and reliability [16]. Those behavioral outputs (like time spent or level of activity) may be disjointed from a behavioral goal that requires the layering of experience and skill to accomplish. For example, auditory and olfactory enrichment is a relatively well-accepted practice [17], but the outcome typically does not align with the adaptive relevance of a particular species’ olfactory or auditory abilities. Essentially, the adaptive relevance of smell or hearing is to acquire resources or avoid threats, which may require a complex suite of behaviors following the stimuli to accomplish—none of which typically follow the provision of olfactory or auditory enrichment in the traditional sense, as most practices lean toward the therapeutic treatment of undesirable behaviors, or the animals simply doing ‘something’ [17], as opposed to triggering a suite of adaptively relevant behaviors with a measurable goal. A hyena is adapted to use its senses to detect an opportunity to feed, then must find that opportunity, perhaps compete for the resource, and process it, all while maintaining vigilance. That suite of behaviors following a sensory input from the environment can be measured as successful or not, adding value to the behavior that doesn’t exist with simply the provisioning of a scent. 

The temporal and spatial context in which animals exist in managed care has been thoughtfully evaluated, specifically in the case of providing resources (food, enrichment, etc.) that are balanced between predictable and random locations or time points [18]. Research and theory have also focused heavily on the concept of “the 24-h” lives of animals in managed care [19], arguing that many husbandry practices are incompatible with the natural rhythm of most species in human care. Brando et al. [19] goes on to cite and discuss the breadth and complexity of the natural history of managed species, with carefully thought-out comparisons to the most common temporal needs, including appropriate sleeping periods and normal feeding times. This theoretical approach coupled with the validated impact on animal welfare these practices can provide is accurately categorized as what they ‘experience’ in human care. However, that is only when we functionally use the term ‘experience’ to describe an input. If we eliminate the constraint offered by traditional (input-based and acute) enrichment, we can develop strategies that shift the meaning of the term experience from an input to a cycle of learning and the acquisition of skills.

### What Is Experience 

Cambridge Dictionary defines experience, when used as a noun, as “(the process of getting) knowledge or skill from doing, seeing, or feeling things” [20]. Certainly, it is this process that we aim to influence in a traditional enrichment program, yet the practice is almost blindly employed with the assumption that the animal already knows how to respond; we just have to elicit it. Sheperdson [21] offers one of the more widely accepted descriptions of the value of enhancing environments in line with a natural behavioral repertoire to increase an individual’s behavioral choices and encourage species-appropriate behaviors and innate skills. As cogent as this definition is, it does not truly address the subsequent value of developing these abilities or the complex sequence of behaviors (layering) that must follow for them to match behavioral choices in line with the natural history of a particular species. The layering of behaviors adds significant complexity, as it requires an understanding of how a specific sequence of behaviors can result in a benefit to the individual and create a more dynamic relationship with the environment. When we borrow a theory from human learning, Social Cognitive Theory (SCT) [22], we can see the cascade of components founded in experience that control the relationship between individual, environment, behavior, and acquisition (or avoidance). SCT prescribes that an individual’s ability to acquire and use knowledge is influenced by experiences they have accumulated by observing the environment and the results of how one interacts with it. One of the primary tenets of SCT is in the framework called Reciprocal Determinism [23]. Reciprocal Determinism follows a stepwise process by which effective learning is developed: The interaction of a person with a set of learned experiences and the environment in which they find themselves, resulting in a behavioral response that results in success [24]. If we apply this model of learning to animals in managed care, we can easily understand that the value of a reciprocal relationship between an animal and its environment results in behavior as a response to the stimuli. This is familiar to many practitioners, as it involves the same basic principles that function with operant conditioning, an almost universal practice in zoos. The striking difference when applying this learning theory to wildlife care is the source of the stimulus. Unlike training in a traditional sense, experience building relies on stimulus from the environment and not directly from the caretaker. If we are to strictly adhere to this model, we must also understand that a dynamic and reciprocal relationship is dependent on a set of learned experiences. It is this experience that forms a more adaptively relevant set of experiences (events that one experiences) for the individual. 

If we circle back to the idea that wild animals face numerous challenges, which they are equipped to deal with based on natural history and physical and cognitive adaptations [11], and are also part of a temporal cycle that does not mirror their human caretakers [19], we can start to see the fundamental gaps in our attempts to provide for their experience. By adding context to both the idea of being equipped for challenge and the time between those challenges being dynamic, we can see that the need for a behavioral response (even at random intervals) must still be triggered by a stimulus. The nature of traditional husbandry limits the variation in that stimulus and inadvertently blunts the quality and quantity of experiences needed to respond to stimuli beyond the existing variation. It is our opinion that traditional care programs can inadvertently limit the value of inputs that are used to accumulate knowledge by providing those inputs in ways that don’t match the individuals’ sensory skills. These sensory skills are used to make determinations about how to interact with one’s environment and have evolved as critical learning tools. As an example, a wild black kite (*Milvus migrans*) in Kenya can sense the impending rainstorm (individual with learned experience). Knowing the rain causes termites to swarm (environment, external social context), the bird heads to the nearest termite mound in anticipation of a meal (behavioral response to stimuli). The challenge in the above situation is knowing that rain will come and change the environment; the experience of knowing both that and where the nearest termite mound is located allows the bird to exploit the resource. The temporal nature is mostly random, as the bird or the termites may be active at any point, but the cue that triggers the cascade is clear and reliable. It will only rain if the environmental cues exist (perhaps barometric pressure change), the termites will only be accessible if it rains, and the bird must know that to achieve its behavioral goals. It seems simplistic, but when compared to the experience of a bird in managed care it becomes obvious that most of that mechanism is missing.

Leaning again on Brando [19], we can explore an even deeper level of this sequence of events and the permutations that coincide with the overall life history of an individual animal. When we apply this multi-step framework to a change in season that alters access to resources, we can start to glean the actual length of time between the environmental cue and the final step. Using the example of a frugivorous species, the individual must follow the cycle of the fruit, both temporally and spatially. Perhaps not all edible fruits develop at the same rate, and the ability to distinguish between a plant that is budding, flowering, fruiting, or out of season presumably takes experience to determine. This perspective would appear to require a concept of how both time and location work in relation to acquiring resources and making determinations based on inputs—namely, the developmental stage of the resources one hopes to acquire in comparison to another resource and the behavior that follows. This is also a process that takes substantial time on the part of the developing fruit and patience for the frugivore. If we try to compare this cycle to that of a traditional captive frugivore, it does not take extensive evaluation to determine that both the experience required and the variability in behavior are fundamentally different. We would be hard-pressed to withhold a favored resource for a week or more, despite the value it adds to the animal’s ability to gain that experience, simply because it does not align with what some accreditation organizations describe as a modern zoo practice [25]. This is even more relevant in pre-release conservation programs, in which ‘modern’ zoo practices may hinder the success of a release if animals lack the appropriate skills to survive a dynamic natural environment [26]. The tools and strategies of Outcome-Based Husbandry seek to make these vital connections by supporting learning and experience that lead animals to develop skills that allow them to understand the dynamics of environmental changes. Ideally, those skills are fostered by providing environments with real life relationships between how the environment changes and how (to be successful) an individual must change as well. 

## 3. Outcome-Based Husbandry

In a heavily critical, but spot-on, assessment of the standard approaches to measuring zoo animal welfare, Watters et al. [27] delivers a cogent blow to the notion of husbandry based on the exact wild environment of a species. What the authors describe is certainly the most common flaw in the idea that husbandry should be based on facilitating the expression of natural behavior as it would happen in a strict sense of that individual’s actual wild environment. Provisioning of input-based enrichment, “naturalistic” enclosures, and an otherwise cursory approach to measuring behavior are, in most cases, missing the mark with regards to the value of that behavior. The flaw in this logic is we can underestimate the motivation and complexity of a suite of adaptive behaviors because we are trying to replicate the superficial nature of the wild environment, without focusing on the relationship one has with the environment. To take this even further, how those behaviors develop and maintain plasticity has almost no place in traditional husbandry because there is no need to, say, change one’s foraging goals because it’s raining. A healthy food (like what has been provided all year) will be presented at a predictable time, regardless of the barometric pressure or relative humidity. Those stimuli (environmental cues) have no value to the captive animal because its experience tells it that there is no need to alter its behavior. Relevance of a behavior is the target of Outcome-Based Husbandry (OBH) and appropriate responses to stimuli learned over time can provide that relevance. Again, application of an SCT type model, developing experiences that span time, space, season, and life stage may hold the key to eliminating the description of ‘natural behavior’ as the foundational goal of traditional enrichment. This evolution of traditional husbandry may also require the retirement of the traditional concept of enrichment, and the input-driven trappings it holds as tenants. Animal welfare was (until recently) guided by the input-based principles outlined in the original Five Freedoms [28] until being rendered obsolete by outcome-based principles like the Opportunities to thrive [29], and the Five Domains [30]. It is the authors’ opinion that enrichment, too, has reached its zenith under the foundational S.P.I.D.E.R model [7] and can now be rendered obsolete as an input-based supplement to an otherwise incomplete husbandry approach. An Outcome-Based Husbandry program, by contrast, uses an ethologically based workflow tool to incorporate adaptively relevant behaviors and experiences into a dynamic program that no longer relies on enrichment to supplement a complete husbandry program.

### Building the Model

The outcome-based workflow tool that starts with potential behaviors and identifies expected outcomes drives the OBH model. Modeling experiences to develop an OBH program need not be complicated or resource-dependent but does have to be equal parts applicable and collaborative. Not unlike a traditional enrichment program, OBH begins with a species and a stated behavioral goal (See Table 1 for an example of modeling an OBH program). However, unlike enrichment, this goal should be viewed as a broader categorical reference to a behavioral suite as opposed to a specific or perhaps one-off behavior. The goal is to develop skills that can be layered to adapt to changing inputs along a variable cycle. For example, “foraging” can be facilitated by the provision of any number of puzzle-type feeders, but ethologically it is irrelevant because it simply requires manipulation to complete. Furthering this example, the cue is the presence of the puzzle, the challenge is to manipulate the object, and the outcome is the acquisition of food. One does not need to rely on variable experience to determine the best way to acquire the resource; simple trial and error might be all it takes. It misses the opportunity to truly explore the depth of the behavioral suite in which perhaps weather, competition, distribution, physical prowess, sensory inputs, season, motivation, and problem-solving are all critical to accomplish the “natural behavior”. For most species, foraging is a multidimensional process that involves any number of physical and cognitive approaches, all shaped by the pressure of ecological and environmental factors. In a traditional system, the puzzle feeder is supplemented as enrichment, where many of the resources would be provided in a predictable non-contextual way (feed hopper, dish, platform, etc.). The goal in OBH is that all provisions of resources are dependent on a complex suite of environmental cues and opportunities for decision-making requiring relevant motivation. 

Once a behavior is determined, the context (Table 1) of those goals can be thought out, paying attention to the variability that the behavioral suite encompasses. In addition to the specific food item, flooding, drought, clumped resources, social competition, suspended, buried, need for processing, or highly challenging access are all components of a foraging process. All these variations in context also require a physically and cognitively distinct approach (on the part of the individual animal) and require experience in reading the ecological cues to understand which approach is most likely to succeed. The practitioner is then free to determine which physical, cognitive, or sensory adaptations (Table 1) are required for each one of the different contexts. Those adaptations should be relevant to both the goal and the context, which will provide us with the insight we can use later when determining how to signal the animals as to which tools from its experience base it needs to employ. For example, visual acuity and physical strength are needed to determine the distance and what relative body posture will yield the most favorable resources when foraging in the terminal branches of a tree. 

Establishing a viable list of outcomes (Table 1) is the way a practitioner can determine if the animal met the criteria for a behavioral response relevant to the context and using the predicted adaptations. From a practical standpoint, outcomes can also serve as a viable record to meet regulatory standards. The outcomes must be measurable to form the foundation of the husbandry practice. The outcomes also embody the variation that the OBH model has established thus far. Again, this is in line with SCT, in which the individual-environment-response complex dramatically enhances the variability and sequence of each experience. The diversity of behavioral outcomes must remain relevant in terms of frequency and direction to avoid the pitfalls often associated with a single behavioral measure [27,31]. 

The final step is to establish the inputs (Table 1) that align with the first four columns to create experiences (events) that allow for the acquisition of more experience (knowledge) for the animal we are dealing with. Inputs challenge the practitioner to formulate the husbandry program in a way that conforms to the flow of the OBH table and enables a series of adaptable experiences that can replace the traditional day-to-day approach. Altering one context can change the adaptation and outcome, facilitating a change in input. The input is also the primary place for one to explore the temporal conditions that differentiate between context, adaptation, or outcome. The input column is also possibly the most challenging, as it eschews traditional husbandry and may produce an experience that foregoes any traditional enrichment for a period of time, something some practitioners may be reluctant to adhere to. The example of the developmental stages of a resource like fruit is a viable and relevant experience within the context of the natural history of an animal. However, a thoughtless implementation of it (such as providing unripe fruit with no context or experience), coupled with an uninformed audience, can be misinterpreted as a failure to meet ‘modern zoo practices.’ Through acceptance of new methods of care and culture change, we can influence not only public perceptions but also how regulatory bodies assess standards of care and welfare. 

## 4. Enriched Events Provide Experience

Once the OBH model is understood and multiple tactics have been developed, it is possible to start building layered events that occur over days, weeks, months, or years. The idea is that we replace standard husbandry (like food provision) supplemented by enrichment with a storyboard of events that match an ecological pattern. In doing so, we can eliminate the need for a traditional enrichment program to supplement an otherwise static husbandry program. These events must include meaningful sensory cues, which are ultimately paired with a predictable input culminating in the relevant behavioral outcome. Animals in managed care experience their world according to the inputs chosen for them, such as habitat design, feeding regimens, enrichment, and conspecifics, further depreciating the value of responding to the environment in which they live. Operational needs, time management, and conservation directives may influence these inputs, but the proportion of that influence needs to be assessed critically. However, by concentrating on promoting outcomes that are meaningful, we can provide more purposeful inputs. Providing meaningful cues that are honest and reliable allows the animals in managed care the opportunity to make biologically relevant choices [32] based on environmental cues that should supersede unreliable cues like the presence or absence of caretakers. When these cues are part of the layered events, a life story can develop where what happened yesterday impacts today and what happens today will affect tomorrow. 

Developing events that stitch together builds knowledge and subsequently increases the animal’s capacity to make meaningful choices, which is the main motivation for replacing enrichment with Outcome-Based Husbandry to enrich the animals’ experience. From the standpoint of the practitioner, this is as simple as replacing traditional husbandry with a more relevant approach. Again, events are multi-day experiences that use sensory cues to strengthen the relationship of the animal with its environment. Sensory cues are used to signal potential inputs that will occur over the course of an appropriate time and provide meaningful inputs that an individual processes to choose a preferred outcome. Those inputs, in turn, become reliable cues as to what might occur in the following days. For example, a very common species/fact combination in modern zoos is that of the serval’s (*Leptailurus serval*) ability to jump up to 9 feet straight into the air to catch a bird in mid-flight. Time of day, brush covering, and other environmental cues likely play a role in determining the area with the best potential to catch a bird, all of which are already addressed by modern zoo design and management. The choice to “hunt birds” and the skills required to do it are seldom found as a standard part of the normal exhibition of the species. A simplified example of how a modern-day zoological facility could use sensory cues and inputs to elicit an event based on their natural ability to capture and process birds can be found in Table 2.

It is important to note that the inputs are not a replica of “natural history,” as it is the choice of behavior(s) informed by experience that we are looking to foster. In this example, the food item does not have to be a bird. It could be a typical diet item with molted feathers pressed into it, subsequently requiring feather plucking behavior before consumption. This event adds complexity and a narrative that runs for several days and builds an experience base for how to manage this challenge in the future. As with any cue, it is crucial that it be reliable and consistent. Repeating events strengthens the cues and allows the animal to plan for opportunities that may occur in the proceeding days. Every time it rains, and the insects come out, there is an opportunity to catch a “bird.” Every time it rains, and there are no insects, there will not be a “bird,” but perhaps a different opportunity. Layering these events to create a relevant sequence of experiences replaces the traditional provisioning of food and supplemental enrichment in an adaptive way that highlights the frequency and diversity of behavior as a validation of progress.

Of course, the roadblocks hindering any advancement in a field as nuanced as animals managed in captivity are neither negligible nor insurmountable. Of primary concern to most practitioners (pers comm.) is the rigidity of schedules and the lack of time for complex husbandry adjustments. Admittedly, the process outlined here does have a time cost, but it is mitigated by its position in the sequence of development. We describe the process as heavily front-loaded with respect to a time commitment, with an eventual efficiency reached once a “cultural change” in husbandry procedures is normalized. The inherent thought process that accompanies building an OBH program can easily replace complex enrichment construction and supplementation, as the events themselves do not adhere to a traditional object input paradigm. The documentation requirements have also been cited as a source of anxiety (pers comm.), with various accreditation and certification bodies requiring some form of effective tracking [13,25]. However, the front-loaded nature of the process also means that the measurable outcomes have already been determined and documented when one designs an OBH program. The question of the efficacy of the outcomes becomes a simple yes-or-no proposition, as they have already been predicted in the steps of the process. Further, the acute measurements of activity or engagement no longer maintain the same relevance if the object is to create a more comprehensive experience for the animal. In this case, measures of behavioral diversity are extremely effective for monitoring longitudinal change. Behavioral diversity indices alone do have shortcomings [31], but when appropriately used can track the richness of a behavioral repertoire as well as the proportion (frequency) of individual behaviors that are expressed. 

## 5. Conclusions 

The expansive development of captive animal enrichment has played a major role in the advancement of animals in human care, but it has presumably reached its culmination in both efficacy and application. Husbandry advancements have reached a tipping point, in which the idea of having to supplement it feels almost like an admission of failure. Recognizing and respecting the capacities of the animals in our care forces us to reconsider both how we got to this point and how we plan to make a difference in the future of the field. The natural history argument is flawed [27], not because of its intention, but rather its execution. We should not seek to mimic an animal’s natural history in a modern zoo due to what it purports, not least of which is an unrealistic expectation coupled with ineffective measurability. Shifting our paradigm to one more in line with capacity building ultimately decreases the inherent dependence on human actions to a more “natural” system of animals adapting to human influence. We can foster the cognitive ability to match physical adaptations with environmental changes in a much more relevant way. We must heavily rely on the vast literature from not only our field but others to nurture adaptable programs with measurable outcomes that ostensibly destroy the “check the box” principles we have been trained to depend on. Providing animals with the appropriate inputs to build their experience bolsters their capacity to manage a complex world of experiences and allows us to retire the concept of supplemental enrichment by building an Outcome-Based Husbandry program that allows for dynamic and relevant experiences daily. 

## Figures and Tables

**Table 1 animals-12-01293-t001:** Species—Corvid, Behavior—Foraging (consider as a suite).

**Context** *Break down behavior into various contexts or components*	**Adaptations** *List adaptations that allow the animal to execute behavior according to context/component.*	**Outcomes** *Measurable outcomes you would expect to see if successful at eliciting the behavior within the context.*	**Inputs** *A layered approach to what can be implemented guided by the previous categories.*
Terminal branches	Visual acuity, physical strength, alertness (anti-predator), spatial awareness, temporal awareness, beak, balance, taste, wings, strong legs	Suspensory feeding, appropriate response to displacement (threat), increased competition when clumped, decreased latency to find suitable resources, avoidance of unsuitable resources, removal of obstacles, increased waste distribution, increased exploration, increased habitat use, consistent response to environmental cues, increased foraging efficiency, good feather/body condition, improved problem solving skills.	Resources only provisioned on terminal branches with varying degrees of difficulty. Seasonal variation in distribution (abundance, clumped, dispersed). Variation in developmental stage of resources depending on environmental condition (following rain, thaw heatwave). Signaled following an abundance of pollinators or other arthropods specific to each resource item. Sent or visual cue unique to each type of resource item, preceding provisioning in a reliable manner.
On stalks			
Following other species			
On ground			

**Table 2 animals-12-01293-t002:** Serval enriched event (over 3 days).

Serval	Day 1	Day 2	Day 3
Natural Cue	“Rain”	“Insects appear after storm”	“Birds appear to eat insects”
Cues as inputs	Naturally occurring rain or misters left on all day as if it were a storm	Live crickets released from slow feeders.	Food item hung at top pf habitat, must jump to obtain reward.

## Data Availability

Not applicable.

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
