# Peer review of "Enrichment Is Simple, That’s the Problem: Using Outcome-Based Husbandry to Shift from Enrichment to Experience"

_animals, 2022, doi:10.3390/ani12101293_

Round 1

Reviewer 1 Report

I believe that the article was well written. It could be useful to improve the current model of enrichment environment applied to captive animals in order to their wellbeing. 

Author Response

Thank you

Reviewer 2 Report

Thank you so much for that brilliant article and your thoughts.

Author Response

Thank you, we look forward to advancing these concepts.

Reviewer 3 Report

This commentary outlines a shift from traditional enrichment practices to Outcome-Based Husbandry. I believe the ideas presented here are certainly worth publishing, but I feel the manuscript, as written, needs substantial revision.

In particular, I feel that to make this a really valuable contribution to the literature you need to:

  • simplify the language so that it is accessible to non-experts (zookeepers and other non-scientific experts)
  • Increase the amount of time you spend explaining the OBH model and provide many more examples about how this could be practically implemented. Once I really understood what you meant (after the paragraph at line 300) I wanted to know much more about HOW to do this. This would likely necessitate a reduction in the amount of space you spend on the introduction

Below are my specific comments by line number:

55-57: While you include references here, it would be nice to give some examples of what you are talking about instead of making the reader lookup those references. This seems crucial, since the rest of the paragraph depends on the reader knowing what you are talking about.

Throughout – the language used is very dense and technical and low frequency wording is used when plain language would work just as well if not better. As this paper should be targeted to those working in zoo settings in addition to scientific experts, I think the paper would benefit from revisions to make the paper easier to read.

Examples:

69: “Theoretically, this bearing may be fostered by the…” could be more simply stated as “This may be due to the…”

75: “in so much that the regulation clearly states” could be “stating..”

86: “express behaviors is noble and ubiquitous in compliance documents” could be more simply stated as “Such definitions are common in compliance documents.”

112: This sentence would also benefit from being simplified.

These are just a few examples, but I really think this could be a much stronger paper if it were reworked for clarity and simplicity since ideally non-scientists will be reading it.

96-97-  An example of an inappropriate species-specific behavior here would add clarity.

101 – I think you should define what you mean by behavioral outputs. It would probably help to define what you mean by inputs as well.

119 – An example here would also be helpful.

This paper uses a lot of direct quotes from other sources. While direct quotes certainly have their place in scientific writing, that is usually limited to situations in which the specific voice of the author needs to be heard or the exact wording is essential. I feel that the use of direct quotes in this paper could be reduced or possibly eliminated.

135 – A description of why layering would be a better standard would be nice

137 – A brief definition of SCT would be beneficial.

151 – Extra space between sentences

156 – I don’t think you need to redefine experiences, the part in parentheses

166 – “It is our opinion that a driving force behind shunting inputs that develop the knowledge necessary for adaptively relevant behaviors is the undervaluing of sensory modalities that have evolved as critical learning tools.” This is a dense sentence. Rewrite for clarify.

175 – I doubt that rainstorms are “entirely random”

Example starting in line 168: This is really the strongest writing to this point in the paper. You give a clear example and relate it to your point. I would like to see you do this throughout. In fact, I might move this example much earlier in the paper (or add other examples earlier in the paper) that support your arguments this clearly.

198 – Can you explain what you are referring to when you say modern zoo practice? I am not sure what practice you are referring to. Also, would this not be akin to fasting some carnivores, which is done at modern zoos?

201 – Extra space between sentences

198- 201 – Don’t most pre-release programs do some sort of training for release? I am thinking specifically of free-ranging golden lion tamarins here prior to their release into the wild.

203 – Why do animals need to be competent and confident? I don’t disagree with you, but your logic here could be better explained.

207-210 – This could be explained a bit better? What the actual flaw is and why it is a flaw is not clear.

214 – Don’t most zoos vary diet, particularly in regard to fresh fruits and vegetables based on what is seasonally available?

Paragraph at 205 – It may be worth mentioning, somewhere in this paragraph, that natural behavior can also negatively impact an animal’s welfare. The stress that comes with being hunted, for example, is not something we would want to replicate in captivity.

227 – A description of the SPIDER model is warranted. Again, if you want non-experts (i.e., zoo-keepers) to read this, you need to not rely on them looking up every reference.

229 – You finally define OBH – I would move this a bit earlier in the paragraph. Also, you need to explain what you mean by no longer needing enrichment. Animals in captivity need to be provided with things that increase their mental stimulation. Even if we switch to a different mode of enrichment, this is something that needs to be done. To say you are going to do away with enrichment needs a lot more support and explanation.

242 – What do you mean it is ethologically irrelevant? If a species is a manipulative forager, how is getting them to express that behavior not relevant to their behavioral ecology? Granted it may not be as complex as determining the season and then deciding how to forage, but it is still providing mental stimulation.

246 – Trial and error – this really depends on the complexity of the puzzle feeder.

253-255 – Ok, so how would you practically do this in a captive setting?

279 – Extra space in between sentences

Paragraph at 256 – Ok, so it kind of seems like you are saying we need to make enrichment in captivity more reflective of the way behaviors would emerge in a natural environment. I agree that adding complexity would be great. But, earlier in the paper you also say that our goal should not be to facilitate natural behavior – yet it seems like that is what you are describing – a way to add more complexity so that animals behavior in captivity better reflects the behavioral chains that we would see in the wild…

292-294 – I agree with you statement, however, providing unripe or over-ripe fruits may also be toxic to animals. I don’t think we necessarily need them to learn this to thrive in captivity…

315 – Presence or absence of a keeper is an extremely reliable cue of when an animal will be fed. What do you mean here?

Paragraph at 300 – I feel like I am just not beginning to see your point. I think you spend too much time criticizing current practices and not enough time really getting into the details of your proposed OBH model. However, I still wonder how this would practically be implemented at a zoo given the realistic constrains on keeper time, food availability, seasonality, weather events, etc. There would be so much variation that I am having trouble conceptualizing how a zoo would plan something like what foods they would order for their commissary that week. If all resources were available all the time, this seems like it would be possible, but they aren’t. We, humans, have predictable work schedules and animals in managed care will pick up on those same rhythms to some extent.

342 – Extra space

337 – Good example in this paragraph. I would like to see more of this.

Author Response

We would like to thank this referee for the very thorough and thoughtful review. These recommendations are very helpful and provide needed input. This overall assessment is very clear and perhaps we can make the language more accessible based on some of your recommendations. The non-expert is not the intended audience but rather the facilitators of the larger programmatic approaches in the zoo and aquarium industry. We have been teaching these techniques for over five years now, and the practitioners understand how this works. It is the director level staff that need to help drive these universal changes in the industry. Our intention is to present the framework of this paradigm shift on a wider scale to set a platform for critical improvements to the basics of animal husbandry, within the literature and in practice. This manuscript is intended to focus on the “why” we need a shift, with some “how” we do it. Subsequent manuscripts are meant to really dig into the how’s in specific conditions, but are difficult without the foundations established prior. Our responses are in italics.  

This commentary outlines a shift from traditional enrichment practices to Outcome-Based Husbandry. I believe the ideas presented here are certainly worth publishing, but I feel the manuscript, as written, needs substantial revision.

In particular, I feel that to make this a really valuable contribution to the literature you need to:

  • simplify the language so that it is accessible to non-experts (zookeepers and other non-scientific experts)
  • Increase the amount of time you spend explaining the OBH model and provide many more examples about how this could be practically implemented. Once I really understood what you meant (after the paragraph at line 300) I wanted to know much more about HOW to do this. This would likely necessitate a reduction in the amount of space you spend on the introduction

Below are my specific comments by line number:

55-57: While you include references here, it would be nice to give some examples of what you are talking about instead of making the reader lookup those references. This seems crucial, since the rest of the paragraph depends on the reader knowing what you are talking about.

Thank you, we have added some more language as you suggest.

Throughout – the language used is very dense and technical and low frequency wording is used when plain language would work just as well if not better. As this paper should be targeted to those working in zoo settings in addition to scientific experts, I think the paper would benefit from revisions to make the paper easier to read.

We will try to simplify some of the language, but this seems more of a style correction rather than accessibility. Again, in our experience most practitioners understand this process clearly.

Examples:

69: “Theoretically, this bearing may be fostered by the…” could be more simply stated as “This may be due to the…”

This is a great edit and we have applied it

75: “in so much that the regulation clearly states” could be “stating..”

Thank you, we have used your suggestion

86: “express behaviors is noble and ubiquitous in compliance documents” could be more simply stated as “Such definitions are common in compliance documents.”

We have used your suggestion

112: This sentence would also benefit from being simplified.

We have re-written this sentence to be simpler.

These are just a few examples, but I really think this could be a much stronger paper if it were reworked for clarity and simplicity since ideally non-scientists will be reading it.

96-97-  An example of an inappropriate species-specific behavior here would add clarity.

We have added an example

101 – I think you should define what you mean by behavioral outputs. It would probably help to define what you mean by inputs as well.

We have defined what we are talking about based on the traditional outputs we see in most enrichment programs.

119 – An example here would also be helpful.

We have provided an example

This paper uses a lot of direct quotes from other sources. While direct quotes certainly have their place in scientific writing, that is usually limited to situations in which the specific voice of the author needs to be heard or the exact wording is essential. I feel that the use of direct quotes in this paper could be reduced or possibly eliminated.

We have eliminated all the direct quotes except the USDA regulations and the Cambridge definition, as we believe the specific voice and exact wording are essential.

135 – A description of why layering would be a better standard would be nice

We have added a justification

137 – A brief definition of SCT would be beneficial.

Thank you, this makes more sense and helps the section

151 – Extra space between sentences

Corrected

156 – I don’t think you need to redefine experiences, the part in parentheses

Removed

166 – “It is our opinion that a driving force behind shunting inputs that develop the knowledge necessary for adaptively relevant behaviors is the undervaluing of sensory modalities that have evolved as critical learning tools.” This is a dense sentence. Rewrite for clarify.

We have re-written this sentence for simplicity.

175 – I doubt that rainstorms are “entirely random”

Entirely has been replaced with mostly

Example starting in line 168: This is really the strongest writing to this point in the paper. You give a clear example and relate it to your point. I would like to see you do this throughout. In fact, I might move this example much earlier in the paper (or add other examples earlier in the paper) that support your arguments this clearly.

We felt it was appropriate to set up this example, as in practice we have had to establish the foundations to this point before most people can grasp the impact of this example.

198 – Can you explain what you are referring to when you say modern zoo practice? I am not sure what practice you are referring to. Also, would this not be akin to fasting some carnivores, which is done at modern zoos?

We added some relevance to the term for clarity and in practice, withholding a favored resource is not akin to a single day of fasting for carnivores in this context.

201 – Extra space between sentences

198- 201 – Don’t most pre-release programs do some sort of training for release? I am thinking specifically of free-ranging golden lion tamarins here prior to their release into the wild.

Some do, but most focus on things like predator avoidance and less so on the dynamics of resource acquisition. This is why so many still provide feed to released animals.

203 – Why do animals need to be competent and confident? I don’t disagree with you, but your logic here could be better explained.

We agree and have tried to make it clearer.

207-210 – This could be explained a bit better? What the actual flaw is and why it is a flaw is not clear.

We have tried to clarify this

214 – Don’t most zoos vary diet, particularly in regard to fresh fruits and vegetables based on what is seasonally available?

Certainly, some do, but our point is not the actual fruit rather the differences in accessing and processing that would change through the seasons.

Paragraph at 205 – It may be worth mentioning, somewhere in this paragraph, that natural behavior can also negatively impact an animal’s welfare. The stress that comes with being hunted, for example, is not something we would want to replicate in captivity.

We have included a reference to this with your previous comments about line 96-97

227 – A description of the SPIDER model is warranted. Again, if you want non-experts (i.e., zoo-keepers) to read this, you need to not rely on them looking up every reference.

We added some language to fix this.

229 – You finally define OBH – I would move this a bit earlier in the paragraph. Also, you need to explain what you mean by no longer needing enrichment. Animals in captivity need to be provided with things that increase their mental stimulation. Even if we switch to a different mode of enrichment, this is something that needs to be done. To say you are going to do away with enrichment needs a lot more support and explanation.

We have softened the language to align more closely to the proposition of changing a husbandry approach.

242 – What do you mean it is ethologically irrelevant? If a species is a manipulative forager, how is getting them to express that behavior not relevant to their behavioral ecology? Granted it may not be as complex as determining the season and then deciding how to forage, but it is still providing mental stimulation.

That is accurate, but we are advocating for the complexity of the context you describe to further the mental stimulation provided by manipulating something.

246 – Trial and error – this really depends on the complexity of the puzzle feeder.

Thank you, the language has been changed to acknowledge that fact.

253-255 – Ok, so how would you practically do this in a captive setting?

Table 1 is the actual tool we use to develop these strategies.

279 – Extra space in between sentences

Paragraph at 256 – Ok, so it kind of seems like you are saying we need to make enrichment in captivity more reflective of the way behaviors would emerge in a natural environment. I agree that adding complexity would be great. But, earlier in the paper you also say that our goal should not be to facilitate natural behavior – yet it seems like that is what you are describing – a way to add more complexity so that animal’s behavior in captivity better reflects the behavioral chains that we would see in the wild…

We addressed this in the other correction based on what you recommended in lines 207-210 to clarify what is often understood as natural behavior.

292-294 – I agree with you statement, however, providing unripe or over-ripe fruits may also be toxic to animals. I don’t think we necessarily need them to learn this to thrive in captivity…

I have added some language to make it clear we are not advocating for hazardous activities.

315 – Presence or absence of a keeper is an extremely reliable cue of when an animal will be fed. What do you mean here?

It’s a quite unreliable cue. The presence of a keeper can mean several potential outcomes (fed, cleaned, moved, darted, observed, etc…) but the cue is always the same (the presence of the keeper). The cue does not explicitly inform the animal of the subsequent outcome.

Paragraph at 300 – I feel like I am just not beginning to see your point. I think you spend too much time criticizing current practices and not enough time really getting into the details of your proposed OBH model. However, I still wonder how this would practically be implemented at a zoo given the realistic constrains on keeper time, food availability, seasonality, weather events, etc. There would be so much variation that I am having trouble conceptualizing how a zoo would plan something like what foods they would order for their commissary that week. If all resources were available all the time, this seems like it would be possible, but they aren’t. We, humans, have predictable work schedules and animals in managed care will pick up on those same rhythms to some extent.

Again, both tables are actual procedures that we have used in multiple instances and facilities. We are attempting to present a fundamental change in the way we approach animals care and recognize that a paradigm shift of this magnitude takes steps. The first step in developing OBH programs in different captive care situations has been to introduce the conceptual framework first and then problem solve the specifics of the actual organization. In some case this takes monumental shifts in culture regarding scheduling and husbandry philosophy, in others its just takes a nudge. Our intention is to get the foundation into the literature to foster the development of programs across the board.

342 – Extra space

337 – Good example in this paragraph. I would like to see more of this.

Reviewer 4 Report

The paper is a commentary that addresses the issues relating to environmental enrichment for animals kept in artificial environments. The authors propose an Outcome-based Husbandry, using an ethologically based workflow tool with a universal apllication.

The paper clearly explains this new approach and highlights its advantages. In my opinion, the paper can be accepted for publication.

I congratulate the authors on their excellent work which will provide useful information to those who deal with animals in captivity and will allow a reformulation of the concept of environmental enrichment, improving animal welfare.

Author Response

Thank you for your review.
